# Light Control in Microbial Systems

**DOI:** 10.3390/ijms25074001

**Published:** 2024-04-03

**Authors:** Yara Elahi, Matthew Arthur Barrington Baker

**Affiliations:** School of Biotechnology and Biomolecular Science, UNSW Sydney, Kensington 2052, Australia

**Keywords:** light control, bacteria, photoreceptors, LOV, microfluidics, motility

## Abstract

Light is a key environmental component influencing many biological processes, particularly in prokaryotes such as archaea and bacteria. Light control techniques have revolutionized precise manipulation at molecular and cellular levels in recent years. Bacteria, with adaptability and genetic tractability, are promising candidates for light control studies. This review investigates the mechanisms underlying light activation in bacteria and discusses recent advancements focusing on light control methods and techniques for controlling bacteria. We delve into the mechanisms by which bacteria sense and transduce light signals, including engineered photoreceptors and light-sensitive actuators, and various strategies employed to modulate gene expression, protein function, and bacterial motility. Furthermore, we highlight recent developments in light-integrated methods of controlling microbial responses, such as upconversion nanoparticles and optical tweezers, which can enhance the spatial and temporal control of bacteria and open new horizons for biomedical applications.

## 1. Introduction

Light dependence is a common phenomenon in various species’ development, behavior, and physiology. Light dependencies encompass vision, photomorphogenesis, phototropism, and phototaxis across diverse species [1,2]. Beyond its energetic significance, light plays an important role in enhancing perception and environmental cognition, while also controlling the synchronization of biological cycles, which regulates the organization of physiological activities. Recent research implied that light can also modulate key pathogenic and virulence factors in ESKAPE pathogens [3].

Light’s properties and its significant impact on organisms have prompted scientists to seek approaches to control biological mechanisms using light. Apart from existing methods, which are based on chemical genetics, photochemical induction, and electrophysiology, the emergence of optogenetics can push back the boundaries of controlling cellular and molecular processes with light. Harnessing optical methods for controlling bacteria is an exciting prospect in biotechnology and synthetic biology, such as developing nanomotors that might alter medication delivery systems. Considering significant advances, researchers continue to develop highly efficient techniques. Later in this review, we explore various methods of light activation, comparing their respective strengths and weaknesses, and discuss recent advancements in light control.

## 2. Light Activation in Bacteria

Bacteria use sensory proteins [4] and RNAs [5] to detect particular intracellular and extracellular signals, which are then used to set up a regulatory response. Photosensitive proteins function as optical sensors or actuators. They either provide a fluorescent signal to identify changes in biological activity or allow light to control cellular biological operations [6,7]. A range of bacterial and eukaryotic light sensors have been developed to detect ultraviolet, blue, green, red, and near-infrared signals (Table 1).

At the molecular level, light is detected by sensory photoreceptor proteins, which perceive light and convert it into biochemical responses that can sense different parts of the electromagnetic spectrum, spanning from near-ultraviolet (near-UV) to near-infrared (NIR) wavelengths. In other words, sensory photoreceptors play a role in converting the absorbed photons into biological changes by their chromophore, which can include alterations in enzymatic activity or interactions with other biomacromolecules [23]. Some of the photoreceptors are specific to prokaryotes, while others are also found in eukaryotes. For example, the predominant light-sensing components in *Escherichia coli* that have been examined include phytochromes and LOV (light–oxygen–voltage) proteins. Figure 1 illustrates various types of photoreceptors and their light-responsive behavior.

Photoreceptors are categorized according to their chromophore and specific photosensory protein domain. These chromoprotein photoreceptors are responsible for perceiving light signals and translating them into biochemical signaling pathways that trigger physiological responses within the cell.

### 2.1. Photoreceptors

Organisms have developed specialized photoreceptive proteins (photoreceptors) for regulating cellular responses to light [24]. Photoreceptors have a unique dual-state functioning, switching from an inactive state to an active one in response to light exposure. This adaptability allows organisms to adjust to changing light circumstances and maintain homeostasis regardless of environmental obstacles.

Across different evolutionary groupings of species, photoreceptors maintain a strong genetic relationship [25] and exhibit comparable responses to bilin and flavin chromophores, which are engaged in important biological activities such as light sensing in plants [26]. Molecules such as sensory rhodopsins (SRs), phytochromes, photoactive yellow protein (PYP), phototropin (LOV), and blue-light sensing using FAD (BLUF) can function as light-sensing agents capable of modifying organismal behavior and life cycle (Figure 2) [20].

SRI is a key receptor that helps microorganisms move towards or away from light, a behavior known as phototaxis. It was the first receptor discovered to sense light in microorganisms and was found during studies on *Halobacterium salinarum* movement in response to light [28,29]. Within a one-photon photochemical reaction cycle, a signaling conformer of the protein accumulates as a long-lived (~800 ms) spectrally shifted intermediate in SRI. A second photon excitation of the molecule efficiently photoconverts the photochemically reactive signaling conformer back to the unphotolyzed (or “dark”) state in approximately 70 ms. Because of the photochromic interaction between 1-photon formation and 2-photon reversion, the organism is able to distinguish between colors through color-sensitive signaling and phototaxis [30].

All phytochromes are structurally categorized into three subfamilies based on the number of domains in the photosensory core module. Three domains, namely, PAS (Per-ARNT-Sim), GAF (cGMP phosphodiesterase-adenylate cyclase-FhlA), and PHY (phytochrome-specific domain), have been identified in the photosensory core module of members of the most common “canonical” phytochrome subfamily. These domains have distinct amino acid sequences; however, they have comparable structural topologies [31]. Cyanobacterial phytochromes (Cph), which do not have an N-terminal PAS domain, and cyanobacteriochromes (CBCRs), which do have a single GAF domain, are another two subfamilies [32].

BphPs are bacterial photoreceptors that detect red and far-red light, orchestrating diverse physiological responses to these lower-energy wavelengths. LOV domains have been found in bacterial genomes [33] and belong to the PAS (Per-Arnt-Sim) domain superfamily, which share an evolutionary connection with HK (histidine kinase) and helix-turn-helix (HTH) DNA binding domains. LOV domains need a flavin cofactor (flavin adenine dinucleotide (FAD)) or flavin mononucleotide protein (FMN)) and are linked to a variety of signaling output domains [34] and control activities such as general stress response [35], cell envelope physiology [36], and virulence [37].

PYP was first discovered in the halophilic purple bacteria *Halorhodospira halophila* and has a possible role as a light sensor in negative phototaxis [38]. PYP, or photoactive yellow protein, utilizes p-coumaric acid (pCA) as its chromophore for sensing blue light, with a peak absorbance at 446 nm. When exposed to blue light, pCA undergoes trans-to-cis isomerization, leading to the formation of a red-shifted state known as pR. In this state, Glu46 transfers a proton to pCA, inducing structural changes and partial unfolding of the protein. This transition results in the formation of a blue-shifted state referred to as pB, which is believed to serve as the signaling state enabling PYP to communicate with its interacting partners. Subsequently, pCA undergoes isomerization to return to its initial ground state (pG), thereby completing the reversible photocycle of PYP [39].

The phototropin blue light receptors (phots) are proteins that have made significant contributions to both plant physiology and protein engineering [40]. LOV domains are found in phototropins and bind the FMN in response to blue light exposure, inducing a green fluorescence in the bacteria [41].

BLUF photoreceptors have a variety of roles, including phototaxis, enzyme photoregulation, and photosynthetic gene regulation [42]. BLUF photoreceptors bind to an oxidized FAD chromophore that can absorb UV-A and blue light. When activated by light, the FAD absorption shifts slightly towards red by 10–15 nm, suggesting that the FAD stays oxidized. A rearrangement of hydrogen bonds around the FAD is likely the mechanism behind this activation [43].

### 2.2. Chromophores

Like other organisms, bacterial photoreceptors need a chromophore to function. In general, proteins that require a cofactor for function are called apoenzymes or apoproteins, and in light-sensitive systems the cofactor is typically a chromophore and the apoprotein a protein such as an opsin (see below). Chromophores are molecules responsible for light absorption; thus, chromoprotein photoreceptors are responsible for perceiving light signals and translating them into biochemical signaling pathways that trigger physiological responses within the cell.

Most recognized chromophores, such as FMN, FAD, p-coumaric acid, keto-carotenoids, and CYG (cysteine-tyrosine-glycine), primarily absorb light in the blue spectrum (400–500 nm). In contrast, a few, like retinal, coenzyme B12, or 5’-deoxy adenosylcobalamin (AdoCbl), have a wider absorption range, ranging from ultraviolet (UV) to green and blue (300–570 nm). Linear tetrapyrroles like biliverdin and phycocyanobilin are sensitive to red/far-red light (620–750 nm) [44].

FMN, which is a chromophore for LOV proteins, is initially noncovalently linked to the protein. Upon absorbing blue light, FMN undergoes a transformation into a triplet state, prompting a covalent bond with a conserved cysteine residue. This covalent attachment signifies the active (signaling) state of the photoreceptor. Subsequently, through thermal processes, the covalent form reverts to the protein’s dark state [45]. On the other hand, BLUF domains use FAD chromophores, which is non-covalently bound within a mixed α/β fold structure for blue light detection [46].

Rhodopsin, phytochromes, and xanthopsins utilize a photochemical mechanism that relies on the E/Z isomerization of their associated chromophores retinal, phytochromobilin, and p-coumaric acid, respectively, while cryptochromes and proteins with LOV or BLUF domains use flavin molecules for their photochemical responses [47].

### 2.3. Opsins

A chromophore interacts with opsin proteins, which can be categorized into two clusters: type I, present in bacteria, archaea, and eukarya, and type II, exclusive to animals [48,49]. However, findings from a practical study imply that type I and type II opsins descended from a single ancestor [50].

All opsins found in bacteria and archaea are categorized as type I opsins and serve diverse purposes, including light-driven outward translocating H^+^ pumps like bacteriorhodopsin (BR), proteorhodopsin (PR), xanthorhodopsin-like rhodopsin (XLR), light-driven inward translocating Cl^−^ pumps exemplified by halorhodopsin (HR), and light-activated signal transducers (sensory rhodopsin I and II) [30,31,32,33].

Scientists found BR and HR in halophilic archaea [51]. Until the discovery of PR inside marine bacteria, it was thought that non-eukaryotic opsins were restricted to salt-dwelling archaea [52]. Subsequent surveys of marine bacteria and archaea that did not rely on typical culture methods found that genes related to opsins are dispersed across taxonomic groupings and regions. These genes are found in many microbial species that live on the ocean’s surface [53]. Through comprehensive genome analyses alongside investigations into biochemistry and physiology, it was shown that certain marine bacteria possess an additional and recently uncovered role for opsins (an outward pumping of Na^+^ ions referred to as NaR) [54]. Subsequently, opsin-like proteins were identified within eukaryotic microorganisms, encompassing algae and fungi [55].

#### Rhodopsins

Rhodopsins are light-sensitive proteins consisting of opsin apoproteins, which covalently bond to a distinctive chromophore known as retinylidene Schiff base that originates from vitamin A and forms a covalent bond with a conserved lysine residue located in the seventh helix (TM7) [56]. These interactions lead to photon absorption for energy conversion or the initiation of intra- or intercellular signaling. 

While microbial and animal rhodopsins have a similar 7-transmembrane α-helical structure and use a retinal molecule as a chromophore [30], microbial rhodopsins have a broader range of apoprotein molecular structures compared to animal rhodopsins. Microbial rhodopsins comprise all-trans retinal via the protonated Schiff base linkage with a Lys residue positioned at helix G in the dark to absorb visible light [57].

## 3. Controlling Gene Expression and Protein Function with Light-Activated Units

Bacterial genes are DNA functional units that impose their actions using a diffusible product. Genes interact uniquely, based on when and where they are active, giving rise to complex networks. To understand how a single gene operates in a complex system, influences signaling pathways, or affects cells and tissues, it is necessary to be able to regulate its expression precisely. For this purpose, light could be an ideal tool because it can be focused and patterned easily [58].

Light can cause conformational changes and biological responses through the energy pumped in by exposure to light, which moves an equilibrium. Biological responses can be accomplished through two main approaches. The first approach involves chemical modification using photosensitive groups, such as sensory photoreceptors and chemical effectors, such as chelators and isomers. Various photoreceptors, including phytochromes, have been used to regulate gene expression. For instance, phytochrome protein B (PhyB) and phytochrome-interacting factor 3 (PIF3), which dimerize in response to red light and dissociate in response to far-red light, were the first employed photoreceptors for controlling the gene expression [59]. These receptors have been applied in vitro to precisely control when and where bacterial genes are active. Light-sensitive receptors have also been employed to understand and manage the systems that govern gene activity and metabolic processes in bacteria, as documented in various studies [60,61,62,63].

Using photoremovable protecting groups (PPGs) or photocages is one of the most versatile techniques for light activation. These photocages feature a photolabile group that inactivates a biomolecule. Upon exposure to light, the photolabile group undergoes cleavage, thereby liberating the biomolecule to resume its natural function [24]. Chelators can also achieve light control in cells by modulating the intracellular concentration of free metal ions. This method was examined successfully, particularly in developing buffers and optical indicators [24]. Chelator molecules create a “cavity” through the steric arrangement of the carboxylate groups [64]. The caged compound absorbs UV light, leading to the photolysis of the chelator into products with reduced Ca^2+^ affinity [24].

The second approach is light control (Figure 3), a rapidly developing technique, which can control biological processes using light [23,65]. By inserting light-sensitive proteins into cells [66], precise regulation of gene expression and cellular functions, as well as temporal and spatial control, can be accomplished through this technique [67,68,69].

The two-component light-responsive system (TCS) is a mechanism that empowers bacteria to adjust to various environmental changes and can detect a broad spectrum of light, including ultraviolet (UirS/UirR) [70], blue (YF1/FixJ) [71], green (CcaS/CcaR) [69,72], red (Cph8/OmpR) [68], and near-infrared light [73]. The TCSs detect and transduce diverse chemical and physical inputs to initiate appropriate cellular responses [74].

TCSs comprise a sensor histidine kinase (SHK) and a response regulator (RR). SHKs are frequently multidomain signaling proteins organized in a modular architecture that undergo an allosteric transition between two signaling states: active with autophosphorylation and phosphotransferase activity, and inactive with phosphatase activity [74,75]. Through exposure to light, these systems trigger a change in the cofactor-bound HK, leading to kinase activation or deactivation adjustment. This signal is then transmitted via a phosphate group to the corresponding intracellular receptor, which regulates gene expression at the appropriate promoter site (Figure 4) [76].

One-component light sensors provide a way to directly influence protein activity without going through transcription. In bacterial cells, these sensors are typically a part of the blue light-responsive LOV protein family, where an LOV domain is connected to various actuators [76]. In this system, the signal transducer and activator of the transcription domain (STAT) are activated by blue light, which in turn activates the sigma factor and transcription activation (Figure 4) [77,78,79].

## 4. Using Light in Other Ways: Indirect Methods to Couple Light Signal to Microbial Responses

### 4.1. Light-Responsive Nanoparticles

Upconversion nanoparticles (UCNPs) have captivated interest in the light activation of bacteria because of their capacity to convert near-infrared (NIR) light into visible light [80], and the possible activation of light-sensitive proteins or enzymes in bacteria. Light can be utilized to target specific particles, retransmitting a localized signal at a different wavelength. In the context of in vivo applications, UCNPs enhance the non-invasive delivery of visible light into deep tissues, which could benefit tumor therapy and neuroscience. These UCNPs efficiently deliver light to various photomedicines by converting near-infrared (NIR) into visible light, facilitating downstream photo-responsive cellular manipulations [81]. In other words, UCNPs can effectively transform external NIR light into localized blue light, allowing for the non-invasive activation of blue-light-responsive modules in engineered live biotherapeutic products (LBPs) [81].

### 4.2. Optical Traps

Light can also be used directly to apply a force using a technique known as optical trapping, and this has a long history in the study of the bacterial flagellar motor (BFM). Optical trapping stands out as a versatile option among the existing approaches for capturing motile bacteria due to its non-intrusive and accurate characteristics. This method has enabled various findings about bacteria, including their swimming patterns, chemotaxis, and cellular mechanics [82,83,84]. The optical trapping typically relies on a focused laser, often called optical tweezers [85]. Optical tweezers have been traditionally used in the study of bacterial motor biophysics [86]. Optical tweezers use a laser beam through a microscope’s objective lens to capture, manipulate, and exert precisely measured forces on tiny objects with refractive properties [86]. Recently, scientists used magnetic tweezers to accurately measure stall torque in *E. coli* and precisely calibrate the tweezers’ torsional stiffness. Subsequently, motor regeneration experiments were conducted under stall conditions, allowing for the precise determination of stall torque in each torque-generating unit (stator unit) [87].

## 5. How Bacteria Respond to Light: Microbial Response to Light and Engineering Light Control of Bacterial Swimmers

Bacteria can be divided into motile and non-motile groups. Motile bacteria such as *E. coli*, *Pseudomonas aeruginosa*, and *Helicobacter pylori* count on their motility for various purposes. Motility is an important function for the virulence potential of microorganisms [88]. The ultimate advantage of bacterial motility lies in its capacity to enhance a cell’s efficiency in acquiring vital resources within a competitive environment [89]. Prokaryotic cells display a variety of motility types. They may swim through fluid environments by pulling themselves or swarm, glide, or twitch on surfaces, float via air vesicles, and even grab eukaryotic motility machinery, as seen in parasitic prokaryotes [90].

### 5.1. Flagellar Motility

The organ that enables most bacteria to move is called the flagellum. The BFM is characterized as a pair of rotating nano-rings that interact with each other and consist of various proteins [91,92,93], featuring at least three structural components: a basal body, hook, and filament [94,95]. The flagellum uses its rotational motor, fueled by an electrochemical gradient of ions such as protons (H^+^) and sodium ions (Na^+^) across the membrane. The archaeal flagellum is a distinctive motility structure, differing from the bacterial flagellum in composition, assembly, and the utilization of ATP as its power source [96]. 

### 5.2. Chemotaxis: Bacterial Response to Chemical Gradients

Bacterial motility can influence numerous cellular processes, such as chemotactic migration and biofilm development [97]. Chemotaxis involves cells moving towards areas with a higher concentration of an attractant. In the case of eukaryotic cells such as human neutrophils, they can directly perceive and respond to differences in attractant concentration in the environment. This contrasts with bacteria, which mainly rely on detecting changes in attractant levels over time to carry out chemotaxis [98,99]. One challenge in this field is the complexity and multifaceted nature of how cells respond to chemoattractant stimulation [100]. Regulating bacterial motility in response to external stimuli holds the potential for advancing sections like biosensor technology [101,102], collective behavior patterning at a population level [101,103,104], and even applications like targeted delivery agents [101].

### 5.3. Phototaxis: Bacterial Response to Light

Phototaxis refers to the behavior in which an organism adjusts its movement in reaction to light. A photoreceptor captures light and transforms it into a biological signal; a chain of signal transmission then influences the cell’s motility machinery to result in modification in the organism’s movement [105]. The light source can be adjusted for light wavelengths, brightness, and direction over time, allowing for stimuli pattern creation [106]. Different light wavelengths lead to various responses. These include slower movement of colonies in response to red and far-red light [107], while blue and ultraviolet (UV) light and high light conditions prompt negative phototaxis [108]. The ability to connect light perception with motility control has been observed in a wide range of prokaryotes, implying that it must give a range of physiological advantages [109,110].

### 5.4. Engineered Light Responses in Bacteria

The movement patterns of bacterial cells can be controlled by modifying light-responsive proteins. Bacteriorhodopsin can pump ions and transfer them across the cell membrane when exposed to light. If the translocated ion is H^+^, the generated ion gradient helps create the proton motive force (PMF) for ATP synthesis or driving flagellar rotation. By introducing proteorhodopsin into *E. coli* through genetic modifications, bacteria have been engineered to swim only in response to green light [111]. Modified photoreceptors have also been used to regulate gene function in bacteria in response to various wavelengths of light [70,73,112,113]. As we mentioned earlier, the first light-sensitive receptor is SRI, which was initially named the “slow-cycling rhodopsin” [30]. A single SRI directs the cell towards higher intensities of long-wavelength light, which is essential for photoenergy capture by its light-driven pumps, while also guiding the cell away from near-UV light, which reduces photooxidative damage [30]. This extended signaling duration is a characteristic shared by sensory rhodopsin. For instance, channelrhodopsins display a similar color-discriminating mechanism with slow kinetics, allowing scientists to control the lifespan of the spectrally shifted signaling conformer (the conductive state) through photon excitation. Channelrhodopsins can serve as bistable optical switches in optogenetics, being photoactivated by a specific wavelength of light and swiftly reset to the dark state by the light of a different wavelength [114].

## 6. Limitations in Applying Light Control to Biological Systems

The primary hurdle, particularly in the UV region, is that the light poses risks to all prokaryotes. Many bacteria are susceptible to being damaged by intense light with shorter wavelengths, such as UV and blue light [115,116]. The risk is primarily associated with DNA and protein damage and inhibition of the translation machinery due to light-generated reactive oxygen species [110]. Light penetration through biological tissue has limitations due to biomolecule absorption, so when illuminating a biological sample it is necessary to consider the varying depths to which light of different wavelengths may penetrate [117].

On the other hand, flavin chromophores are easily accessible, but their concentration in the cytosol may not be sufficient to provide fully loaded LOV domains [118]. This limitation impacts signal relay efficiency since apo-LOV proteins are non-functional when unloaded. Another challenge is the system’s dynamic range and leakiness due to the imperfection in the association and dissociation dynamics of the LOV proteins [119].

Upconversion nanoparticles might enable more precise spatial stimulation in biological tissue since NIR photons experience less scattering than visible-range photons. This strategy’s limitation lies in the increased complexity of the resulting composite systems. While upconversion nanoparticles are generally not deemed highly toxic, their dissolution can release fluorides and lanthanides, potentially leading to cytotoxicity [120]. Similarly, photocaged molecules have restricted reversibility and spatial control once their protective group is released. As a result, researchers are exploring new methods, such as using photoactivatable proteins or domains as key components in optogenetic protein engineering.

## 7. Towards Improved and Integrated Light Control

Combining optogenetics with inducible gene expression systems to obtain precise control over light-induced gene expression is a strategy that can enhance light control efficiency [121]. Cell-type-specific promoters can regulate an optogenetic constructs’ expression. This method’s specificity increases because only the targeted cell types create light-sensitive proteins. Selecting an optimal light wavelength can minimize activation off-target. In addition, applying patterned or pulsed light stimulation to adjust the time and duration of activation can further improve effectiveness and reduce off-target effects [122].

Microfluidics offers control over individual or small groups of cells within tiny channels and microfluidic structures have been used to investigate the movement of mammalian cells and various microorganisms such as fungi, algae, and bacteria [123]. Their ability to produce nanoparticles with precise control over the size distribution, shape, and high encapsulation efficiency makes them valuable for various applications [124].

In the field of bacterial taxis, microfluidics has enabled the generation of gradients and offers precise, rapid control over the bacterial environment [125]. Moreover, it facilitates experiments involving attractants and repellents at low concentrations while also providing the means to measure kinetic responses. This, in turn, aids in comprehending the speed at which cells can adapt to changing environmental conditions [125].

Some aspects of microfluidic systems are amenable to be integrated with light; for example, the use of polydimethylsiloxane (PDMS) micropatterning can enhance the effectiveness of delivering excitation light and collecting fluorescence in microfluidic systems [123]. Similarly, modern techniques in optofluidic manipulation allow the precise handling of micro- and nano-sized objects within tiny fluid samples [126]. 

We highlighted advantages and disadvantages of various light control methods that we mentioned in this review in Table 2.

Despite the benefit of light control, translating research to application faces hurdles, including scalability and biocompatibility. Addressing these challenges will require interdisciplinary collaboration and efficient engineering approaches to design effective and safe optogenetic systems. Further studies and clinical trials are needed to validate the efficacy and safety of light control interventions in diverse contexts and to better integrate new techniques that can enable the use of light to control motility.

## 8. Conclusions

The connection between light and microorganisms through the lens of light control techniques presents a paradigm shift in the ability to engineer and control microbial systems. The intersection of biology, optics, and engineering, together with utilizing integrated techniques such as microfluidics, has reduced barriers and provided us a means to orchestrate molecular and cellular processes with spatiotemporal precision. Further development in the light control of bacterial motion could usher in a new era of drug delivery, precision medicine, and sustainable bioprocessing, and such prospects for using light as an advanced and clinical method for the external control of bacteria will be unlocked by ongoing research efforts in this area.

## Figures and Tables

**Figure 1 ijms-25-04001-f001:**
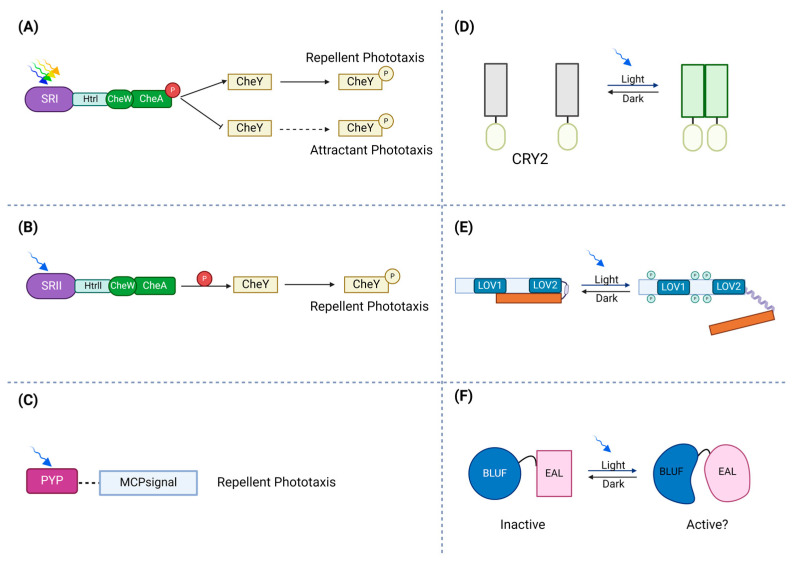
Schematic illustration of photoreceptors’ response to light. (**A**) In the activation of the ion pumps, photo-attractant sensory rhodopsin I (SRI) absorbs light (green to orange range) and causes an attractant phototaxis response. (**B**) To protect against dangerous UV radiation, photo-repellent sensory rhodopsin II (SRII) absorbs blue light and triggers a repellent phototaxis response. (**C**) By attaching to membrane-bound chemoreceptors, which lack sensory domains, PYPs may control phototaxis. (**D**) Transmembrane receptors, cryptochrome 2 (CRY2), can be fused to photosensitive proteins. Via blue light exposure, a conserved N-terminal photolyze homology region of cryptochrome 2 can homo-oligomerize. (**E**) In the dark state, the phototropin receptor is unphosphorylated and inactive. Light absorption by LOV2 disorders the Jα-helix and activates the C-terminal kinase domain, which leads to autophosphorylation of the photoreceptor. (**F**) The absorption of light causes a structural change in the rim enclosing the hook, modifying the protein interface between BLUF and the output domain.

**Figure 2 ijms-25-04001-f002:**
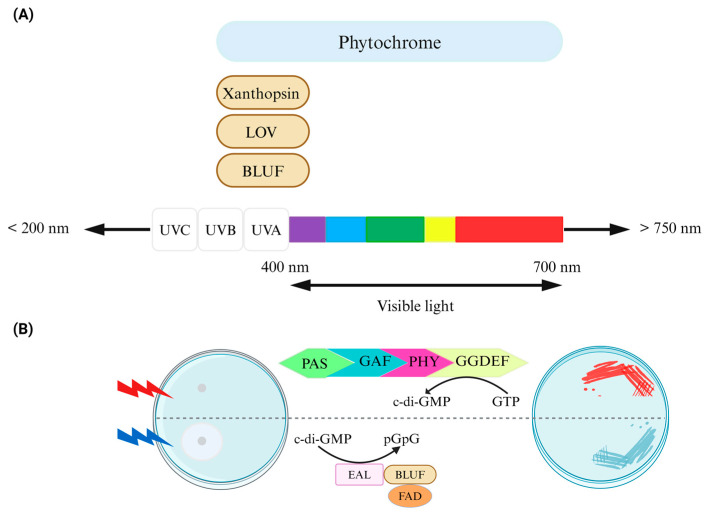
Control of bacterial behavior via light. (**A**) Photoreceptor domains, which are linked to downstream bacterial cyclic dinucleotide signaling domains, exhibit an absorption range. Chromophores such as biliverdin/bilin derivatives, flavin derivatives like FAD, or coumarin, capable of sensing light of varying wavelengths, are either covalently or noncovalently attached to the protein scaffold of photoreceptors in phytochromes (including BphP and cyanobacteriochromes), LOV/BLUF, and xanthopsin proteins, respectively. Phytochrome/phytochrome-like proteins consist of PAS/GAF/PHY domains in diverse combinations. (**B**) The design of an optogenetic system for bidirectional regulation of cyclic di-GMP levels involves a red-light-activated diguanylate cyclase (utilizing a BphP photoreceptor with a biliverdin chromophore) and a blue-light-activated phosphodiesterase (utilizing a BLUF domain photoreceptor with an FAD chromophore) for the control of motility (**left**) and the formation of Congo red-stained biofilms (**right**). Figure adapted from [27].

**Figure 3 ijms-25-04001-f003:**
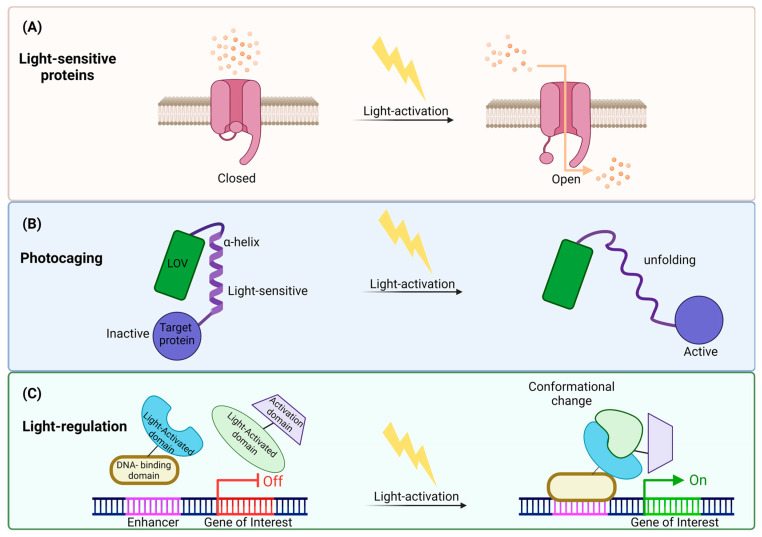
Illustration of light control mechanisms revealing cellular modulation: (**A**) Utilizing light-sensitive channels: One mechanism involves activating light-sensitive channels, opening its gate in response to light. This opening allows the passage of ions across cellular membranes, influencing cellular behavior. (**B**) Light-induced photocaging: Another strategy is light-activation-inducing photocaging, causing structural changes in proteins. The α-helix unfolded and caused light-induced activation in the target protein through this unfolding. (**C**) Conformational changes in regulatory proteins: Light stimulation can also induce conformational changes in regulatory proteins, influencing gene expression. Upon light exposure, the light-activated protein structurally changes and binds to the DNA-binding domain attached to the enhancer. Through this process, transcription is initiated, and the gene is expressed. This illustration underscores the versatility of light control mechanisms in precisely manipulating cellular processes.

**Figure 4 ijms-25-04001-f004:**
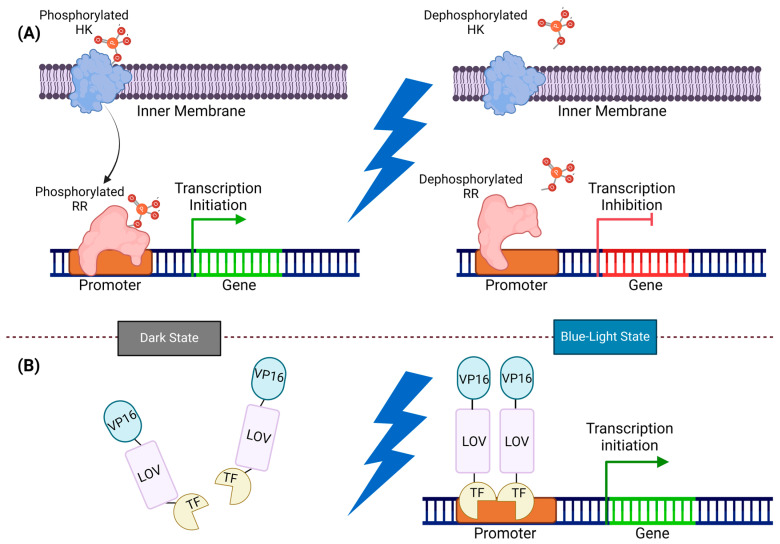
Illustration of two-component and one-component systems: (**A**) In the two-component system, HK and RR work together for signal transmission. In the dark state, HK phosphorylated, transferring the phosphate group to RR, and prompting gene expression. However, exposure to blue light dephosphorylates HK and RR, and inhibits gene expression. (**B**) In the one-component system, LOV monomers undergo dimerization by exposure to blue light and directly initiate gene expression. In the dark state, the VP16-LOV-TF fusion protein remains monomeric, and an inactive LOV domain constrains the TF activator domain. Upon exposure to blue light, the TF activator domain is released from its cage, and the LOV domains form dimers, thereby recruiting transcription machinery by VP16.

**Table 1 ijms-25-04001-t001:** Types of light-interacting proteins and receptors.

Type	Including	Chromophores	Absorbance Range	Ref.
Flavoproteins	MiniSOG, phiLOV	Flavin mononucleotide	~450–520	[8,9]
GFP-like proteins	BFPs, GFPs, RFPs	Tag-BFP like, GFP-like chromophores, DsRed-like	~400–650	[10,11,12]
Bacterial phytochromes	iRFP, IFP1.4, Wi-Phy	Biliverdin	~650–780	[13,14,15]
Flavoproteins	LOV2, CRY2	Flavin mononucleotide/flavin adenine dinucleotide	~440–480	[16,17]
Rhodopsins	Channelrhodopsins, halorhodopsins, OptoXRs	Retinal	~490–630	[18,19,20]
Plant and cyanobacterial phytochromes	PhyB/PIF, Cph1	Phycocyanobilin	~550–740	[21,22]

**Table 2 ijms-25-04001-t002:** Comparative analysis of light control methods in microbial systems.

Methods	Advantages	Disadvantages
Light-sensitive proteins and LOV	Utilizes endogenous photoreceptors and light-sensitive proteins to provide precise spatiotemporal control over biological processes. Provides versatility in chromophore selection and photoreceptor domains, allowing for a wide range of applications.	Necessitates genetic modifications for implementation in specific organisms. Complex signal transduction pathways may impact system efficiency and response kinetics.
Photo-regulation	Enables precise modulation of gene expression and cellular functions using light stimuli. Offers temporal and spatial control, facilitating dynamic manipulation of biological processes. Provides a platform for targeted manipulation of specific cell types and signaling pathways.	Limited application in bacteriology compared to neuroscience and fundamental research. Requires genetic engineering for incorporation of light-sensitive proteins into bacterial cells. Complexities in signal transduction pathways may affect system robustness and reliability.
Nanoparticles and optical tweezers	Offers non-invasive and precise control over microbial responses using external stimuli. Enables deep tissue penetration of light for in vivo applications, while optical traps provide accurate manipulation and study of bacterial motility and mechanics.	Limited reversibility and spatial control with photo-caged molecules post-activation. Complexity in synthesis and optimization of nanoparticles for specific applications. Potential cytotoxicity associated with nanoparticles and photo-caged molecules.
Integrated light with microfluidics	Offers cell-type-specific control over gene expression, enhancing specificity and minimizing off-target effects. Combines optogenetics with inducible gene expression for finer control over light-induced responses. Utilizes microfluidics for precise manipulation of cells and gradients, facilitating studies of bacterial taxis and environmental responses.	Requires precise selection of optimal light wavelengths and patterns to minimize off-target effects and maximize efficiency. Challenges in integrating microfluidic systems with light control mechanisms for seamless operation. Complexity in designing and optimizing integrated light control systems for specific applications.

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
