# Peer review of "Light Control in Microbial Systems"

_ijms, 2024, doi:10.3390/ijms25074001_

Round 1

Reviewer 1 Report

Comments and Suggestions for Authors

The manuscript aims to review an interesting and important subject such as using light to control microbial systems. 

My concerns are the following. It  is not very clear what the paper contributes to the existing academic literature. The subject is already reviewed many times and new insights or a different view on the subject are required to justify a new publication. In the current form the manuscript provides a compilation of basic information from other reviews. The manuscript contains three Figures and all of them are adapted from other reviews (ref 3,10,35)  Who is a potential reader of this review? For a new-comers, it seems to be  not detailed and comprehensive enough; for more experienced scientists, it is  too simplistic. 

Another issue, which is especially important if we are talking about  a review, is  the clarity, accuracy, and completeness of the presentation. My overall impression is that it is not very clear which criteria are used to include  the information in the review; the logical flow of the manuscript is not smooth; the given pieces of information are not always properly connected.

In addition, the manuscript contains a considerable number of inaccuracies and misprints that can be quite confusing

Unfortunately, while the manuscript contains some interesting information, I do not believe that it is  publishable in its current form.

Some misprints:

line 89  “The most well-studied example of type I opsin is rhodopsin, a membrane protein with seven transmembrane domains (PDB code 1U19) that include retinal as the chromophore responsible for light absorption”. Actually, Rhodopsin is responsible for vision in animals and  belongs to type II (but not type I) of opsins. Specifically, PDB code 1U19 is for the bovine visual rhodopsin. 

Lines 81-86:  "Molecules within this category include: sensory rhodopsins (SRs), phytochromes, photoactive yellow protein (PYP), phototropin (LOV), and blue-light sensing using FAD (BLUF) and these function as light-sensing agents capable of modifying organismal behaviour and life cycle (Figure 2) [19]. Based on opsin sequences of amino acids, these proteins can be categorized into two extensive clusters: type I, present in Bacteria, Archaea, and Eukarya, and type II, exclusive to animals [20,21]." The mentioned categorization into type I and type II is performed for rhodopsins, while other proteins (phytochromes, PYP, LOV, BLUF) are from other classes.

Line 247: "Phototaxis consists of a photoreceptor" -- phototaxis is the movement of organisms in response to light stimulus, i.e. photoreceptors are utilized for phototaxis.

Line 255: "Rhodopsin pumps ions and transfers them across the cell membrane when exposed to light." Not all rhodopsins pump ions, the specification rhodopsins types should be added here.

Author Response

Dear Reviewers,

Thank you greatly for your time taken to review our manuscript. We have endeavoured to consider all your suggestions and particularly appreciate the comments around changes to clarity and structure of the review to improve the manuscript. We respond to specific suggestions below and have submitted a tracked changes version of the manuscript indicating specific changes.

Reviewer 1

The manuscript aims to review an interesting and important subject such as using light to control microbial systems. 

My concerns are the following. It is not very clear what the paper contributes to the existing academic literature. The subject is already reviewed many times and new insights or a different view on the subject are required to justify a new publication. In the current form the manuscript provides a compilation of basic information from other reviews. The manuscript contains three Figures and all of them are adapted from other reviews (ref 3,10,35)  Who is a potential reader of this review? For a new-comers, it seems to be  not detailed and comprehensive enough; for more experienced scientists, it is  too simplistic. 

Thanks for your helpful suggestions. We have tried to improve the clarity with respect to the target audience – someone who may not have a background in light activation but be interested in applying these tools or proteins to their system of interest. Regarding the figures we have attempted to make new figures that summarise different approaches in light activation. Wenow have 4 figures:  Figure 2 is redrawn based on previous Figure 2, but Figure 3 and Figure 4 and illustrate types of approaches in light control. We agree, previously there were too many figures adapted from other work and these figures were not pitched quite correctly, some were too specific but thus much material was not covered. We have tried to increase the breadth as a summary and list of tools to help this.

Another issue, which is especially important if we are talking about  a review, is  the clarity, accuracy, and completeness of the presentation. My overall impression is that it is not very clear which criteria are used to include  the information in the review; the logical flow of the manuscript is not smooth; the given pieces of information are not always properly connected.

In addition, the manuscript contains a considerable number of inaccuracies and misprints that can be quite confusing

We have edited errors where indicated (see below), and thank the reviewer for close attention to detail. With regard to structure and logical flow, we have changed the structure of the sections, and in particular, added subsection headings to Section 2, which we feel improves the flow and the grouping of the material. We have also reordered the paragraphs in the taxis section to better connect the material on bacterial motility to the rest of the work.

Unfortunately, while the manuscript contains some interesting information, I do not believe that it is  publishable in its current form.

Some misprints:

line 89  “The most well-studied example of type I opsin is rhodopsin, a membrane protein with seven transmembrane domains (PDB code 1U19) that include retinal as the chromophore responsible for light absorption”. Actually, Rhodopsin is responsible for vision in animals and  belongs to type II (but not type I) of opsins. Specifically, PDB code 1U19 is for the bovine visual rhodopsin. 

We thank the reviewer for this clarification. This was sloppy of us and we have attempted to delete the erroneous material. Specifically, we have written opsin when generally referring to opsins.

Lines 81-86:  "Molecules within this category include: sensory rhodopsins (SRs), phytochromes, photoactive yellow protein (PYP), phototropin (LOV), and blue-light sensing using FAD (BLUF) and these function as light-sensing agents capable of modifying organismal behaviour and life cycle (Figure 2) [19]. Based on opsin sequences of amino acids, these proteins can be categorized into two extensive clusters: type I, present in Bacteria, Archaea, and Eukarya, and type II, exclusive to animals [20,21]." The mentioned categorization into type I and type II is performed for rhodopsins, while other proteins (phytochromes, PYP, LOV, BLUF) are from other classes.

We have removed the material relating to phytochromes, PYP, LOV, BLUF at this point.

Line 247: "Phototaxis consists of a photoreceptor" -- phototaxis is the movement of organisms in response to light stimulus, i.e. photoreceptors are utilized for phototaxis.

We agree. This is also sloppy. We have rewritten Lines 278-289.

Line 255: "Rhodopsin pumps ions and transfers them across the cell membrane when exposed to light." Not all rhodopsins pump ions, the specification rhodopsins types should be added here.

We have adjusted this to specifically refer to bacteriorhodopsin, used to engineer light-activation of bacterial motility.

Reviewer 2 Report

Comments and Suggestions for Authors

On request of IJMS, I have revised the manuscript titled “Light control in microbial systems”, by

Yara Elahi and Matthew Arthur Barrington Baker.

In this paper, Authors have reviewed the relationship between light and bacteria and emphasized the role of light control in modifying bacterial behavior.

General Comments

The topic is interesting, but although at a first reading it seemed to me a field still not sufficiently explored, thus making this review original, by a small survey, I found that several previous works exist reviewing the advances and discoveries in this sector. Anyway, an improvement of the present form could make this paper suitable for publication in IJMS.

Major

Authors should emphasize why this review is original respect those already existing and what are the novelties provided to the audience by this work. What is the information reviewed in this work which enrich the already existing knowledge on the topic? Novelties could also consist in the organization of the work, or in the way by which information has been provided. Accordingly, authors should improve their work by adding proper sentences in abstract, and conclusions. Additionally, a short Introduction section is necessary to provide readers with a proper background, at the end of which, authors should insert sentences highlighting the relevance of the present work.

A criticism of this paper is the total absence of Tables which are essential in a review. I suggest Authors to organize the most suitable information in (at least) five Tables which are reader-friendly modalities to communicate.

The Figures are only three, which is a limited number of images for a review. Many journals require at least five images. Additionally, all the three Figure included in this paper are not original and copied from other works. In my opinion, original images are necessary in a publishable work.

Literature is obsolete. As regards my knowledge, several journals require that in a review at least 50% of references should have been published in the last five years. Here, only 26% of references are dated 2019-2023. Please, update literature in this paper.

A critical discussion by the authors concerning the information that has been provided should be included in the paper.

Conclusions are poor and need improvement.

Comments on the Quality of English Language

Minor editing of English language required

Author Response

Reviewer 2

The topic is interesting, but although at a first reading it seemed to me a field still not sufficiently explored, thus making this review original, by a small survey, I found that several previous works exist reviewing the advances and discoveries in this sector. Anyway, an improvement of the present form could make this paper suitable for publication in IJMS.

We thank the reviewer for their positive suggestions to improve the manuscript.

Authors should emphasize why this review is original respect those already existing and what are the novelties provided to the audience by this work. What is the information reviewed in this work which enrich the already existing knowledge on the topic? Novelties could also consist in the organization of the work, or in the way by which information has been provided. Accordingly, authors should improve their work by adding proper sentences in abstract, and conclusions. Additionally, a short Introduction section is necessary to provide readers with a proper background, at the end of which, authors should insert sentences highlighting the relevance of the present work.

We have added an introduction section to better position the rest of the review and also improved the clarity through the use of subsection headings, as well as rewriting the abstract and early material to guide the reader as to the purpose of this review.

A criticism of this paper is the total absence of Tables which are essential in a review. I suggest Authors to organize the most suitable information in (at least) five Tables which are reader-friendly modalities to communicate.

This was a great suggestion. We have added two tables: 1) a list of light interacting proteins and receptors; 2) a comparison of advantages and disadvantages of specific approaches in light activation. This does organize the material better and allows a quick reader to find the material they want quickly, as well as to discern the pros and cons of further reading, with appropriate references. Thanks for this suggestion.

The Figures are only three, which is a limited number of images for a review. Many journals require at least five images. Additionally, all the three Figure included in this paper are not original and copied from other works. In my opinion, original images are necessary in a publishable work.

We have added additional figures, we now have 4 figures and have drawn all from original sources.

Literature is obsolete. As regards my knowledge, several journals require that in a review at least 50% of references should have been published in the last five years. Here, only 26% of references are dated 2019-2023. Please, update literature in this paper.

We have gone over the material and where possible added more contemporary references to provide more recent examples of research in this area.

A critical discussion by the authors concerning the information that has been provided should be included in the paper.

Conclusions are poor and need improvement.

We have rewritten the conclusions and Sections 5 and 6 on limitations and future integration to better reflect our perspectives on drawbacks and where the field in general is going.

Reviewer 3 Report

Comments and Suggestions for Authors

This review deals with the effect that light has on bacterial systems and how it could be used to control and modify the bacteria behaviour. The review is complete and well written and designed and give a novel vision on bacterial biotechnology. What is missed in my opinion is a clear, simply and short chapter, after the description of the effect of light in bacteria,  explaining, before entering in details,  the possibilities of using light to control/modify/exploit bacterial systems like the movement of cells and other activities that authors describe later in the text. Please add some comments on the possible influence of light on virulence.

I have some minor concerns:

Fig 1 B: please try to explain better this figure

line 31: indicate the meaning of FMN, FAD and CYG

lines 186-194: it seems not to be a bacterial application, please explain

lines 219-228: the description of the structure of flagellum seems not to be needed, could be substituted by a reference

lines 257-258: complete the name of the species H. salinarum

line 303: explain what is microfluidics and how could be applied to the light biotechnology

Some references seems to be old dated like 26, 27, 42, 65, 90...could be deleted or substituted?

Author Response

Reviewer 3

This review deals with the effect that light has on bacterial systems and how it could be used to control and modify the bacteria behaviour. The review is complete and well written and designed and give a novel vision on bacterial biotechnology. What is missed in my opinion is a clear, simply and short chapter, after the description of the effect of light in bacteria,  explaining, before entering in details,  the possibilities of using light to control/modify/exploit bacterial systems like the movement of cells and other activities that authors describe later in the text. Please add some comments on the possible influence of light on virulence.

We have added text to the introduction and specifically some recent research on the role in pathogens.

I have some minor concerns:

Fig 1 B: please try to explain better this figure

We have adjusted the caption to Figure 1B.

line 31: indicate the meaning of FMN, FAD and CYG

We have added full descriptions of these terms.

lines 186-194: it seems not to be a bacterial application, please explain

We have renamed the section headings here to better reflect the material that follows, as well as reordering some sections to try and make sure that bacterial material is in the correct place.

lines 219-228: the description of the structure of flagellum seems not to be needed, could be substituted by a reference

We agree, we have restructured this section so that it is shorter and also in a more appropriate location.

lines 257-258: complete the name of the species H. salinarum

We have included the complete species name.

line 303: explain what is microfluidics and how could be applied to the light biotechnology

We spent some time on this in terms of deciding what to do. We tried adding two additional microfluidics figures, but it was in the end too much material that was taking the review in another direction. We have edited this section for better clarity and flow, and better understanding of exactly which parts of microfluidics we were claiming were relevant to light-activation in bacteria.

Some references seems to be old dated like 26, 27, 42, 65, 90...could be deleted or substituted?

We have updated the references for more contemporary readings where appropriate, as also requested by Reviewer 2.

Round 2

Reviewer 1 Report

Comments and Suggestions for Authors

The authors have significantly improved the manuscript, and corrected the inaccuracies. Authors have also added summarizing tables and new figures, which the reader may find useful for getting a clear picture of the subject. I can suggest only a few minor corrections for the manuscript. 

1. Logical flow problems. 

1.1. Part 2 covers the photoreceptor proteins utilized by bacteria to detect light and generate light response. However, information about light-induced damage seems to be out of context. 

1.2. Line 285 “They ..” seems to be the beginning of a sentence, but no sentence occurs in the future. 

1.3. The logical flow for information presented below Line 285 is not clear. In the “2.1. Chromophore” section the reader expects information about chromophores of photoreceptors. Then, the section “2.2. Apoproteins” is expected. 

However, the authors for some reason consider only LOV and BLUF chromophores in section 2.1. In section “2.2. Rhodopsin” they consider this type of photoreceptors (but not considering the photochemistry of rhodopsins). Finally, section “2.3 Controlling gene expression and protein function with light-activated units” starts with photoremovable protecting groups and seems to be not related with previous two sections. 

Dividing Part 2 into subsections each corresponding to a single class of photoreceptors, considering its photochemistry and functioning in a cell, seems to be a more clear way to present the information here.

2. Figure 1 seems to be not complete. It considers only the photochemistry of LOV and BphPs, and other photoreceptor proteins are missing.

Author Response

We appreciate your feedback on our revised manuscript. We respond below to the first reviewer's additional comments, which required some minor revisions.

  1. Logical flow problems.
  • Part 2 covers the photoreceptor proteins utilized by bacteria to detect light and generate light response. However, information about light-induced damage seems to be out of context.

In part 2, the sentence about light-induced damage has been removed.

1.2. Line 285 “They ..” seems to be the beginning of a sentence, but no sentence occurs in the future.

We have removed this mistake, thank you for catching this.

  • The logical flow for information presented below Line 285 is not clear. In the “2.1. Chromophore” section the reader expects information about chromophores of photoreceptors. Then, the section “2.2. Apoproteins” is expected.

We have revised further the section headings and added information about chromophores to the chromophore section (2.2). We have further defined apoproteins at the top of this section, then provide more detail about opsins as a class of apoprotein below this.

However, the authors for some reason consider only LOV and BLUF chromophores in section 2.1. In section “2.2. Rhodopsin” they consider this type of photoreceptors (but not considering the photochemistry of rhodopsins). Finally, section “2.3 Controlling gene expression and protein function with light-activated units” starts with photoremovable protecting groups and seems to be not related with previous two sections.

We have modified section 2.3 to provide more information about photoreceptors and rhodopsins. We have also changed the structure of different sections and made some changes in section 3 to attempt to better connect these sections. “Controlling gene expression and protein function with light-activated units”.

Dividing Part 2 into subsections each corresponding to a single class of photoreceptors, considering its photochemistry and functioning in a cell, seems to be a more clear way to present the information here.

We have added explanations about different photoreceptors now in Section 2.1.

  1. Figure 1 seems to be not complete. It considers only the photochemistry of LOV and BphPs, and other photoreceptor proteins are missing.

We thank the reviewer for this suggestion. We have now replaced the previously extracted figure with an original figure, which demonstrates more information about different photoreceptors. There is a slight emphasis on chemotaxis adjacent systems but this the system in which we have the most expertise.

Reviewer 2 Report

Comments and Suggestions for Authors

Dear Editors,

I appreciate your revision work, which enables the publication of your paper.

Author Response

We thank Reviewer 2 for their approval.